# Spotlight on the Ballet of Proteins: The Structural Dynamic Properties of Proteins Illuminated by Solution NMR

**DOI:** 10.3390/ijms21051829

**Published:** 2020-03-06

**Authors:** Yuji Tokunaga, Thibault Viennet, Haribabu Arthanari, Koh Takeuchi

**Affiliations:** 1Molecular Profiling Research Center for Drug Discovery, National Institute of Advanced Industrial Science and Technology, Tokyo 135-0064, Japan; tokunaga.y@aist.go.jp; 2Department of Cancer Biology, Dana-Farber Cancer Institute, Boston, MA 02215, USA; Thibault_Viennet@DFCI.HARVARD.EDU; 3Department of Biological Chemistry and Molecular Pharmacology, Harvard Medical School, Boston, MA 02115, USA

**Keywords:** solution NMR, protein function, dynamics, interaction, conformational entropy, drug development

## Abstract

Solution NMR spectroscopy is a unique and powerful technique that has the ability to directly connect the structural dynamics of proteins in physiological conditions to their activity and function. Here, we summarize recent studies in which solution NMR contributed to the discovery of relationships between key dynamic properties of proteins and functional mechanisms in important biological systems. The capacity of NMR to quantify the dynamics of proteins over a range of time scales and to detect lowly populated protein conformations plays a critical role in its power to unveil functional protein dynamics. This analysis of dynamics is not only important for the understanding of biological function, but also in the design of specific ligands for pharmacologically important proteins. Thus, the dynamic view of structure provided by NMR is of importance in both basic and applied biology.

## 1. Introduction

Solution nuclear magnetic resonance (NMR) spectroscopy plays a critical role in providing quantitative information at atomic resolution on the functional dynamics of biomolecules in physiological conditions. NMR can not only be used to determine high-resolution structures of macromolecules, but also to analyze dynamic systems, including the characterization of protein conformational equilibria, conformational changes upon the binding to ligands, tautomeric states, transiently formed interactions, intrinsically disordered regions, and liquid–liquid phase separations. The capacity of NMR to analyze protein and DNA dynamics was described from the very beginning of its application to macromolecules when NMR revealed and characterized the aromatic ring flips in bovine pancreatic trypsin inhibitor (BPTI) [1], permitted by the conformational breathing of the protein. However, since then the development and application of solution NMR techniques have unveiled the functional dynamics of a number of important biological systems, including G protein-coupled receptors (GPCRs), ion channels, and enzymes such as kinases and small GTPases (Figure 1).

In combination with stable isotope labeling, NMR can analyze protein dynamics with atomic resolution without the introduction of any chemical modifications to the protein that could potentially change its dynamic properties. In addition, NMR can uniquely provide access to the lowly populated protein conformations (>0.5%) that are not readily accessible to other structural methods. For this purpose, relaxation dispersion measurements give quantitative information about a relatively slow (μsec to msec) timescale equilibrium. In the Carr–Purcell–Meiboom–Gill (CPMG)-type relaxation dispersion experiment, inversion pulses are applied at an array of frequencies during the fixed transverse relaxation period. A residue that does not undergo conformational exchange will experience one chemical shift and the corresponding intensity will not depend on the CPMG frequency. However, the chemical shift of a residue that experiences conformational exchange will oscillate between the various conformers and hence the intensity of the corresponding resonance will depend on the CMPG frequency. The effective transverse relaxation rate measured as a function of pulsing frequency harbors information about the exchange rate, the population of the states, and the chemical shift difference between the states. Furthermore, quantitative psec to nsec dynamics measurements by NMR allow the estimation of conformational entropy, which essentially reflects the number of the accessible conformations in a macromolecule. Conformational entropy mostly represents the amplitude of the fluctuation of bond vectors in a macromolecule, which can be quantified as squared order parameters by NMR relaxation analyses. NMR-based relaxation experiments can be used to derive order parameters. Thus, NMR plays a crucial role in revealing the essential contribution of the fast-conformational fluctuation of protein in regulating protein–protein affinities and allosteric regulations. Here, we will summarize recent applications of NMR to the dynamics of biologically important systems and how this technique contributes to our understanding of their functional mechanisms and dynamics-guided, structure-based drug developments.

## 2. NMR Unveils Protein Dynamics that Dictate Functions in Key Biological Systems.

### 2.1. G-Protein-Coupled Receptors (GPCRs)

GPCRs are membrane proteins that make up a major class of cellular receptors for external stimuli. GPCRs dictate a plethora of diverse cellular physiological responses, and are therefore an important therapeutic target. Recent breakthroughs in the structure determination of GPCRs and GPCR–transducer complexes by X-ray crystallography have made significant contributions to the understanding of the molecular mechanism of GPCR activation [2]. However, a complete understanding of GPCR activation by various agonists as well as the structural changes that occur upon antagonist binding cannot be elucidated without dynamics information.

Ligand binding to GPCRs often causes structural changes at a distal site on the receptor, resulting in recruitment of G-proteins or arrestins, thus initiating intracellular signaling [3,4]. Many GPCRs demonstrate basal activity and varying levels of activation that are dependent upon ligand efficacy. While full agonists induce maximal activation of the G-protein, partial agonists only induce, as their name suggests, submaximal activation of GPCR signaling. There are also inverse agonists, which reduce receptor activity below the basal levels observed in the absence of ligand. It was NMR that provided the important insight that the ligands can regulate receptor activity by changing the population of each state in the conformational equilibrium between inactive and active structures [5].

The populations of active/inactive conformations, as determined by NMR, were closely correlated with the degree of activation in response to engaging a given ligand [6,7,8,9,10,11,12] (Figure 2A). It is known, for example, that the agonist-bound β_2_-adrenergic receptor (β_2_-AR) does not assume its fully active conformation in the absence of the heterotrimeric G-protein alpha subunit (Gα). Upon Gα binding, the agonist-bound β_2_-AR is driven towards a fully active conformation, tightening the ligand–receptor interaction. Thus, the regulation of conformational equilibrium in the GPCR molecule is bidirectional. It is also interesting to note that under mechanical pressure > 625 bar, the fully active conformation is dominant in agonist-bound β_1_-AR, even in the absence of Gα [13]. This suggests that Gα binding tightens agonist–GPCR interactions, making the ligand binding pocket more compact, which otherwise is relatively loose without Gα bound.

Some GPCRs are coupled to both G-proteins and arrestin pathways. Certain types of ligands, which are called biased ligands, preferentially activate one of the receptor-associated signaling pathways over the other. NMR studies revealed that these GPCRs exist in multiple open conformations and that the relative population of each open conformation determines towards which signaling pathway activation will be biased [14,15,16,17]. In fact, observations of chemical shift perturbations in residues that are sensitive to conformational changes in the transmembrane helices correlated well with the bias factors, which are defined as the ratio of β-arrestin signaling efficacy to G-protein signaling efficacy for each state in the μ-opioid receptor [17].

It should also be noted that NMR detected substantial differences in the functional conformational equilibrium of β_2_-AR with and without T4-Lysozyme fusion to intracellular loop-3, a modification that is widely used in crystal structure determinations of GPCRs [18]. Furthermore, lipid composition has been shown to affect the functional equilibrium of GPCRs and other membrane proteins, something which can be experimentally quantified by NMR using an array of membrane mimetics from detergent micelles to lipid nanodiscs [16,19,20]. Additionally, the buffer conditions in NMR experiments can also be modulated rather freely, even mimicking physiological conditions and triggers, which provides a unique opportunity to dissect the effect of certain ions on the function of membrane proteins [21,22].

GPCRs also regulate the conformational dynamics of the downstream G-proteins. The Gα Ras domain is under a conformational equilibrium between two states. One of these conformational states, the open state, is shown to bind to the GPCR [23]. It has been further shown that Gα rapidly exchanges between ground- and high-energy-state conformations (with tight and reduced GDP affinity, respectively). An oncogenic mutation accelerates GDP dissociation by shifting the equilibrium towards the high-energy conformation [24]. Thus, solution NMR has contributed to the unveiling of protein dynamics that dictate the functions of GPCRs and downstream signaling molecules, including G-proteins, critical to understanding how this biologically essential signaling system would work in normal as well as in diseased cells.

### 2.2. Ion Channels and Other Membrane Proteins

The multi-level activation associated with a change in conformational equilibrium between different functional structures is not unique to GPCRs, but is rather a common feature found in many membrane proteins. Multiple open and closed states with multiple gates are evident even for the simplest ion channel such as KcsA, a prokaryotic potassium channel that contains a single pore domain [25,28,29,30]. In other ion channels, such as P2X, the conformational equilibrium between the open and closed states defines the degree of conductance [20]. In the case of KcsA, NMR successfully characterized the dual gating properties of the potassium channel, where both the opening of the intracellular gate and the activation-coupled inactivation at the selectivity filter are required to have ion conductance and thus both determine the open probability of the channel. Although the electrophysiological open probability of KcsA is rather small (less than 10%), the conformational differences of the intracellular helix bundle (lower gate) between acidic and neutral conditions are such that the gate is fully open at acidic pH (Figure 2B). The discrepancy between the electrophysiological open probability and intracellular gate can be explained by the activation-coupled inactivation at the ion channel’s selectivity filter (upper gate). In the NMR analysis of KcsA, pH-dependent chemical shift perturbations were also observed for the residues involved in a hydrogen bond network related to activation-coupled inactivation. This observation reveals how a structural change in the intracellular gate can be coupled to the distal ion selectivity filter. In fact, NMR was able to separately observe distinctive signals from closed, open, and inactivated conformations by controlling the temperature and pH of KcsA samples, perfectly recapitulating the electrophysiological properties of the channel [29,31]. The NMR analysis also successfully identified a “pH-sensor” for the channel, His-25. Substitution of His-25 by Ala abolished this pH-dependent conformational rearrangement [25].

There are many cases where the molecular basis of membrane protein function can be described as a two-state or multi-state equilibrium [32,33,34]. In some cases, the active form might not be the major conformation. For example, in yeast ADP/ATP carrier 3 (AAC), a protein that is found in the inner mitochondrial membrane, the ^1^H-^15^N transverse relaxation-optimized spectroscopy (TROSY) spectrum shows that AAC in solution dominantly resides in the cytosol-facing conformation. This is conformation is similar to the X-ray structure of the inhibitor-bound state of AAC [35]. However, NMR relaxation dispersion experiments reveal that there is another conformational state populating ~2%. The minor state is assumed to be related to the matrix-facing conformation, since residues that show conformational exchange are localized to the matrix-facing side. In this system, the presence of substrate accelerates the exchange between two conformations, which might contribute to its fast transport [36]. It should be also noted that NMR has contributed to the de novo structure determination of a number of multi-spanning membrane proteins, such as outer membrane proteins (Omps) [37,38,39,40,41,42], voltage-dependent anion channel (VDAC1) [43,44,45,46], mitochondrial uncoupling protein 2 [47], the mitochondrial calcium uniporter [48], the p7 channel from hepatitis C virus [49], diacylglycerol kinase [50], respiratory supercomplex factor 1 (Rcf1) [51], disulfide bonf formation protein B (DsbB) [52], and proteorhodopsin [53].

### 2.3. Small GTPases

The conformation of membrane-associating proteins in contact with the membrane is also uniquely accessible by NMR. The structural and dynamic analysis of the lipid bilayer-anchored small GTPase Kirsten rat sarcoma viral oncogene homolog 4B (K-Ras 4B), and its interactions with effector protein retrovirus-associated DNA sequences (RAS)-binding domains, was investigated by NMR [54]. K-Ras 4B is a membrane-associated GTPase prenylated at the C-terminus and anchored in the plasma membrane. Its membrane localization is essential for function; however, as membrane-association poses a major challenge to crystallization, the structural details of how K-Ras 4B interacts with the membrane were lacking. Structural analysis of K-Ras 4B anchored to lipid nanodiscs doped with spin-labeled lipids revealed that K-Ras is in an equilibrium between two major orientations relative to the membrane, reflective of the activation state of K-Ras 4B. In addition, binding of effector proteins induces a reorientation of K-Ras 4B from the occluded state to an exposed state. Interestingly, gain-of-function mutations, such as G12D, drive K-Ras 4B toward the exposed state in the absence of effector binding, which may partly explain the elevated downstream oncogenic signaling by these mutations. Based on this knowledge, the authors successfully developed a compound with a unique mechanism of action: it stabilizes K-Ras 4B in an orientation that is not suitable for effector-binding, thus impairing activation of downstream signaling [55].

### 2.4. Kinases

Kinases are essential for a variety of cellular processes including signal transduction, transcription, and metabolism. Kinase activity is not only regulated by the substrate and ATP binding, but is often also controlled by post-translational modifications such as phosphorylation, which modulate the dynamics of functionally important sites in the protein. Protein kinases, which represent the largest protein superfamily (consisting of over 500 distinct genes in the human genome), share a conserved catalytic domain that serves for catalysis [56,57,58,59,60,61]. Most protein kinases have N- and C-lobes that contain ATP and substrate binding sites, and the most prominent dynamic movement is a transition from an inactive open to an active closed conformation of these two lobes. Some of the kinases can also adopt pre-activation intermediate conformations, which have partial structural characteristics of the active closed conformation. Although X-ray crystallography has been essential to define the structural basis of protein kinase functions, characterizing functionally important dynamic can sometimes be lost by crystallization. A conformational equilibrium, such as an inter-lobe open/close equilibrium, can be altered to choose the conformation that fits better to the crystal lattice. In addition, a flexible loop, as exemplified by the activation loop in kinases, could also be artificially rigidified by crystal packing, which masks the modular role of the loop in regulating the activity. Furthermore, the indicator of local dynamics in crystal structures, B-factors, can underestimate the dynamics of surface-exposed polar side chains, due to the condensation-enhanced attraction to ions that are supplemented for high quality crystals.

An example of the additional information provided by NMR comes from studies of extracellular signal-regulated kinase 2 (ERK2). The crystal structure of the unphosphorylated/inactive state is open, while that of the phosphorylated/active state is twisted and closed [62,63]. NMR, however, can identify significant dynamics in the μsec–msec time regime between different states of ERK2 [64]. It is known that dual-phosphorylation of the activation loop, which contains the TXY phosphorylation motif (180-TGY-182 for p38α), is necessary to activate mitogen-activated protein kinases (MAPKs). However, in some kinases, the phosphorylation itself may not be sufficient to induce a structural change from inactive to active conformations. There are virtually no spectral changes arising from dual phosphorylation of p38α alone; however, upon the binding of ATP to dually-phosphorylated p38α, larger chemical shift changes were observed [27] (Figure 2C). It should also be noted that almost no spectral change was observed for unphosphorylated p38α, even at a high concentration of ATP, indicating that both ATP binding and dual phosphorylation are essential for induction of the active conformation of p38α [27]. On the contrary, p38γ, another member of p38 subfamily, occupies an intermediate conformation between the inactive and active conformations when unphosphorylated [65]. These examples demonstrate that kinases may have wide ranges of conformational space with distinct distributions in different phosphorylation and ATP binding states. NMR can also contribute to the understanding of how dynamics in different time scales work in concert for kinase function. For example, NMR dynamics studies of adenylate kinase revealed a hierarchy in protein dynamics on multiple timescales that is linked to enzyme catalysis. Atomic fluctuations on the psec–nsec timescale in the hinge regions facilitate large-scale lid motions that result in a catalytically competent state [66,67].

NMR can also identify structural and dynamic interplay amongst distinct functional sites. In studies of the catalytic subunit of protein kinase A (PKA-C), Masterson et al. found that PKA-C cooperatively binds ATP and substrate in the active site, with the pre-loading of either one enhancing the affinity of the other [68]. Analyses of the conformational equilibrium and dynamics of PKA-C by solution NMR elucidated the underlying structural mechanism of this synergy necessary to proceed in the catalytic cycle. Furthermore, in terms of allosteric modulation, NMR studies identified substrate docking interactions at locations distal from the active site that enhance MAP kinase p38α’s activity and substrate selectivity [27]. Allosteric regulation is also active in the intramolecular domain assembly of Abl kinase, mutations in which can lead to oncogenesis [69]. In addition, NMR contributes to understanding the dynamic behavior of kinases in inhibitor-bound states [70,71]. It has been shown that the binding of the ATP-site competitive inhibitor, imatinib, leads to an unexpected detachment of the SH3-SH2 domains from the kinase domain of Abl kinase, which is undone by the addition of the allosteric inhibitor GNF-5, which binds to the C-lobe [72]. The studies provided detailed insights into the combined effect of the ATP-competitive and allosteric inhibitor types, which led to the finding that combination of nilotinib with asciminib resulted in complete control of chronic myeloid leukemia (CML) in a mouse disease model [73].

### 2.5. Enzymes and Other Proteins

Dynamics of proteins are known to contribute to the catalytic function of enzymes. Dihydrofolate reductase (DHFR) is one of the enzymes that has been extensively analyzed by NMR and has become a hallmark system to show the contribution of protein dynamics to catalytic function [74]. In the multistep catalytic cycle, each intermediate has lowly populated high-energy conformations that resemble the low energy structures of preceding and subsequent intermediates [75,76]. Substrate and cofactor exchanges occur through these high-energy conformations, and their binding stabilizes the most relevant conformation for each step, with the energy landscape and the populations of the accessible states changing in response. Furthermore, the reaction velocity and turnover rates are shown to be coupled to the dynamics of the transitions between the ground and high-energy states of the intermediates. Therefore, the energy landscape of the enzyme is dictated by the bound ligands in such a way that the reaction cycle follows a preferred kinetic path between intermediates. The identification of lowly populated (1~2%) high-energy states by NMR is the key to understanding the efficient enzymatic cycle of DHFR.

Analysis of protein dynamics is also important for drug development. CPMG relaxation dispersion measurements showed that the rates of msec-timescale internal motions in the enzyme in each compound’s bound state correlate with the inhibitory constant (*K*_i_) and *k*_off_ for antifolates. This suggests that internal protein motion is critical for ligand dissociation by mechanically initiating the ligand dissociation. As signal transduction, regulatory processes, and pharmaceutical responses are highly dependent upon ligand residence times, the insight obtained in this study that protein dynamics can serve as a mechanical initiator of ligand dissociation has practical importance [77]

While due to space constraints we are not able to discuss in detail the following, it should also be mentioned that NMR has contributed to our understanding of the functional dynamics of a number of enzymes and globular proteins, including chaperones heat shock protein HSP27 [78,79] and 60 kDa chaperonin or GroEL [80], p97 ATPase associated with diverse cellular activities (AAA+) ATPase [81,82], protein tyrosine phosphatase 1B (PTP1B) [83,84], imidazole glycerol phosphate synthase (IGPS) [85], human carbonic anhydrase (HCA) II [86], CRISPR-associated protein 9 (Cas9) HNH nuclease [87], fluoroacetate dehalogenase [88], and multidrug binding transcriptional regulator, quarternary ammonium compounds repressor QacR [89]. The diversity of these examples demonstrates the versatility of NMR to analyze protein dynamics over a wide range of molecular weight as well as timescales.

### 2.6. Intrinsically Disordered Proteins (IDPs)

Intrinsically disordered proteins (IDPs) and disordered regions in proteins have recently been in the spotlight for the functional roles that they play in regulating and fine-tuning key cellular processes. IDPs are difficult to study at an atomic resolution by other structural methods due to their highly dynamic behavior. Even in complex with another protein, IDPs may display high amplitude dynamics from an entirely extended chain to a rather static folding-upon-binding structure. Phosphorylation-dependent folding of IDPs has also been identified by NMR [90]. IDPs are known to engage a target using a number of dynamic and closely related structural conformers termed fuzzy complexes. Dissecting the structure and dynamics of an IDP in its complexed form is therefore of importance to understanding its specificity, affinity, and functions. For example, it has been shown that the intrinsically disordered docking module of the MAPK kinase MKK4 displays significantly different dynamics across the bound region [91]. It has also been shown that histone H1 and its nuclear chaperone prothymosin-α associate in a complex with extremely high picomolar affinity, but both IDPs fully retain their structural disorder and highly dynamic character even in complex [92]. In the competition between the binding of two IDPs, hypoxia inducible factor (HIF)-1α and Cbp/p300 interacting transactivator with Glu/Asp rich carboxy terminal domain 2 (CITED2), to a common target, telomere length regulator TAZ1, the flexibility of the binding motif in the HIF-1α-TAZ1 complex allows formation of a transient ternary complex of CITED2- HIF-1α-TAZ1, which increases the rate of HIF-1α dissociation [93]. This mechanism is thought to allow a rapid and efficient attenuation of the hypoxic response mediated by HIF-1α. Knowledge of the dynamics of an IDP upon binding to a target protein can also be used to improve the affinity of a ligand [94]. DEAD-box helicase 4 (Ddx4) is an IDP that can induce liquid–liquid phase separation to form membrane-less organelles; however, its molecular nature is poorly understood. NMR showed that in the phase-separated state, translational diffusion of Ddx4 along with the small molecules encapsulated in the phase separated environment is 100 times slower than that of Ddx4 in its monomeric form, even though its disordered region remains disordered and highly dynamic. NMR also successfully identified that Ddx4 molecules form a network of interactions upon phase separation, providing a structural dynamic insight into the functional domain [95]. These structurally dynamic behaviors of proteins in phase separated milieus are uniquely accessed by solution state NMR, giving it the potential to reveal underlying mechanisms.

## 3. Importance of Dynamics in Protein Interactions and Drug Development

### 3.1. Conformational Entropy: Role of Fast Dynamics in Recognition

In addition to the slow concerted motions of proteins and the associated conformational changes, the fast fluctuations of side chains are important for function. Specifically, the contribution of conformational entropy to protein interactions has been shown to be significant. Contributions of conformational entropy to the Gibbs free energy can be estimated from experimentally determined order parameters of the fast timescale intramolecular fluctuations [96]. Although it is difficult to define absolute entropies, differences in entropy can be estimated with reasonable assumptions (see reviews [97,98,99,100,101,102]). It has been shown that the conformational entropy of calmodulin estimated from the change in fast methyl dynamics explains the entropic contribution of the interaction and is estimated to be on the order of −10 kcal/mol [102,103]. Although current NMR relaxation analysis techniques can typically be applied to backbone amides or methyl-bearing amino acids, the order parameters of methyl-bearing amino acids serve as a favorable proxy of the entire order parameter including surrounding amino acids. Therefore, changes in the fast dynamics of methyl-bearing amino acids can empirically be utilized as an “entropy meter” for estimating the changes in conformational entropy in multiple protein interaction systems [104,105]. It has also been shown that, in the case of a mutant of a transcription factor catabolite activator protein, CAP, a large conformational entropy gain originating from enhanced protein motions upon binding to DNA makes the protein DNA-binding competent, even though it resides mostly in an inactive conformation and needs to confer a larger energy penalty to adopt the DNA bound conformation [106].

The enhancement of dynamics upon binding to compounds is also considered to contribute to the improvement of ligand affinity [107]. We have shown that promiscuous but high-affinity recognition of a multidrug binding transcriptional repressor is dictated by conformational entropy in the multidrug binding protein lincomycin resistance repressor LmrR, which serves as a sensor for toxic compounds in bacteria [108]. Compounds binding to LmrR resulted in a notable increase in the amplitude of psec–nsec dynamics at the interface between the compound-binding and DNA-binding domains, which points to an allosteric coupling of these sites [108]. The conformational entropy associated with these changes in dynamics corresponded to −2~−3 kcal/mol, thereby favorably contributing to the high affinity but promiscuous binding of the multi-drug binding transcriptional regulator. The compound-binding site of LmrR exhibits μsec–msec dynamics in the *apo* state, and compound ligation shifts this pre-existing conformational equilibrium to varying extents. It should be noted that the conformational entropy gain associated with compound binding shows significant correlation with the extent of the compound-induced changes in the conformational equilibrium (Figure 3A). Therefore, the conformational equilibrium of the protein that allows promiscuous ligand binding is directly coupled to a high affinity interaction via conformational entropy.

### 3.2. Utilization of Dynamics Information for Drug Design

As is already evident in the LmrR case above, the dynamics of proteins as well as ligands are important for drug design. While the conformational dynamics of small molecules and ligands in their receptor-bound states have rarely been investigated, use of dynamics information could be of importance in future drug developments. For example, Lee et al. investigated inhibitors of UDP-3-O-(R-3-hydroxymyristoyl)-N-acetylglucosamine deacetylase (LpxC), a validated novel antibiotic target, by using NMR [109]. From the analysis of ^13^C chemical shifts and ^3^J couplings of the ligand, they found that the inhibitor accesses alternative, minor population states of the ligand in solution in addition to the major conformation observed in crystal structures. The minor-state conformation defined a cryptic inhibitor interaction site on the protein, and a novel inhibitor that utilized the cryptic site was designed to better incorporate the new interaction site. The strategy led to the development of a potent antibiotic with inhibition constants in the single-digit picomolar range and showed improved antibiotic activity by 2- to 25-fold relative to the original compound against a wide range of gram-negative pathogens.

Namanja et al. demonstrated the ability of NMR to conduct a flexibility–activity relationship study [110]. In this study, they use ^13^C relaxation–dispersion measurements leveraging the natural ^13^C abundance to a series of related ligands that target a common receptor, the peptidyl-prolyl isomerase Pin1, and compare the site-specific changes in ligand dynamics upon binding to the receptor [111]. The comparisons revealed how ligand structure can perturb ligand motions important for activity and provided quantitative site-specific information for ligand mobility.

Mizukoshi et al. showed that the conformational flexibility of bound ligands can also be defined by forbidden coherence transfer analysis in free-bound exchanging systems (Ex-FCT), using the interaction between a ligand, a myocyte enhancer factor 2A (MEF2A) docking peptide, and a receptor, p38α, as a model system [94]. In the study, FCT framework was extended to systems under free-bound exchange in order to evaluate the local dynamics and the surface complementarity of weak-affinity ligands in the receptor-bound state. Applying the Ex-FCT method to a ligand bound to perdeuterated receptor gives local psec–nsec dynamics information of methyl groups, whereas the surface complementarity for each methyl in the ligand–receptor interface can be estimated from a set of Ex-FCT experiments that makes use of receptor with different degrees of deuteration. Interestingly, Val-7 γ and Ile-9 δ1, which are located in the binding pocket of p38α, exhibit more dynamics than the solvent-exposed Val-8 γ (Figure 3B). The lower mobility of Val-8 γ on the psec–nsec time scale seemed to originate from the limited rotameric states of the methyl groups due to proximal water (Figure 3B; cyan sphere), which is involved in a hydrogen bond network between p38α and the MEF2A docking peptide. The results revealed that the dynamics of individual methyl groups did not necessarily correlate with that group’s degree of the surface exposure. Interestingly, the Ex-FCT experiment also identified that the surface complementarity of the Val-7 γ2 methyl group is not optimal and that there is extra space surrounding it. Based on that information, Val-7 was substituted with a larger amino acid, Ile, to see if the affinity for p38α would improve. The V7I mutant showed 3.1-fold greater affinity to WT p38α as compared to the WT MEF2A docking peptide. Thus, the information obtained from Ex-FCT experiments can be used to find suitable sites from the point of view of dynamics to introduce modifications for ligand optimization. It should also be noted that the FCT method is applicable to non-labeled ligands as well as ligands containing trifluoromethyl moieties [112].

## 4. Future Perspectives

The versatility of NMR stems from the significant freedom allowed in experimental conditions. One such highlight is in-cell NMR, which allows the analysis of the dynamic behavior of proteins and nucleic acids inside living cells [113,114,115,116,117,118,119]. The application of NMR dynamics information to drug development has also not been limited to conventional modalities. Protein NMR strategies are readily applicable to analyze the structure and dynamics of biologics and their heterogeneities. As biologics are mostly expressed in mammalian cells, and therefore cannot be deuterated to high levels, ^15^N-detected experiments with TROSY selection are an attractive option because ^15^N transverse relaxation is least affected by deuteration [120,121,122], along with the recently developed ^19^F-^13^C TROSY technique for aromatic sidechains [123]. In addition, medium-sized molecules such as cyclic peptides and macrocycles tend to be flexible in solution. Thus, NMR would be a powerful tool for the analysis of their dynamic interactions. The unique information provided by NMR can also be integrated with other structural methods, such as X-ray crystallography, small-angle X-ray and neutron scatterings (SAXS and SANS), and cryo-electron microscopy (cryo-EM), as well as with in silico strategies [124,125,126,127,128,129,130]. In combination, these structural techniques synergistically combine to illuminate the ballet of proteins and help to elucidate their biological function and ways in which that function can be modulated for therapeutic purposes.

## Figures and Tables

**Figure 1 ijms-21-01829-f001:**
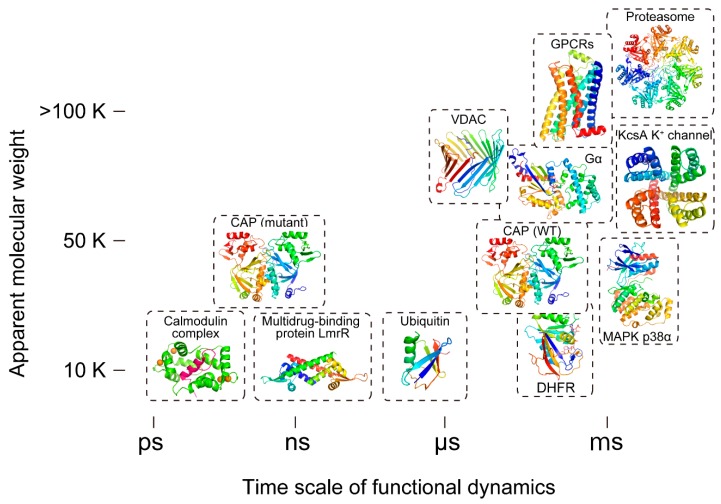
Timescale of functional dynamics and the apparent molecular weights of systems where dynamics were measured using NMR. Time scale of NMR-identified functional dynamics depicted versus the apparent molecular weight of the measured system (inclusive of the micelle or reconstituted high-density lipoprotein (i.e., nanodiscs) in the case of membrane proteins) are shown. It should be noted that the figure highlights only the timescales that have been shown to be essential for functional dynamics in NMR; however, timescales beyond those depicted in the figure may also be functionally relevant. GCPR: G-protein-coupled receptors; VDAC: voltage-dependent anion channel; CAP: catabolite activator protein; Gα: G-protein alpha subunit; KcsA: K channel of streptomyces A; MAPK: mitogen-activated protein kinase; DHFR: dihydrofolate reductase.

**Figure 2 ijms-21-01829-f002:**
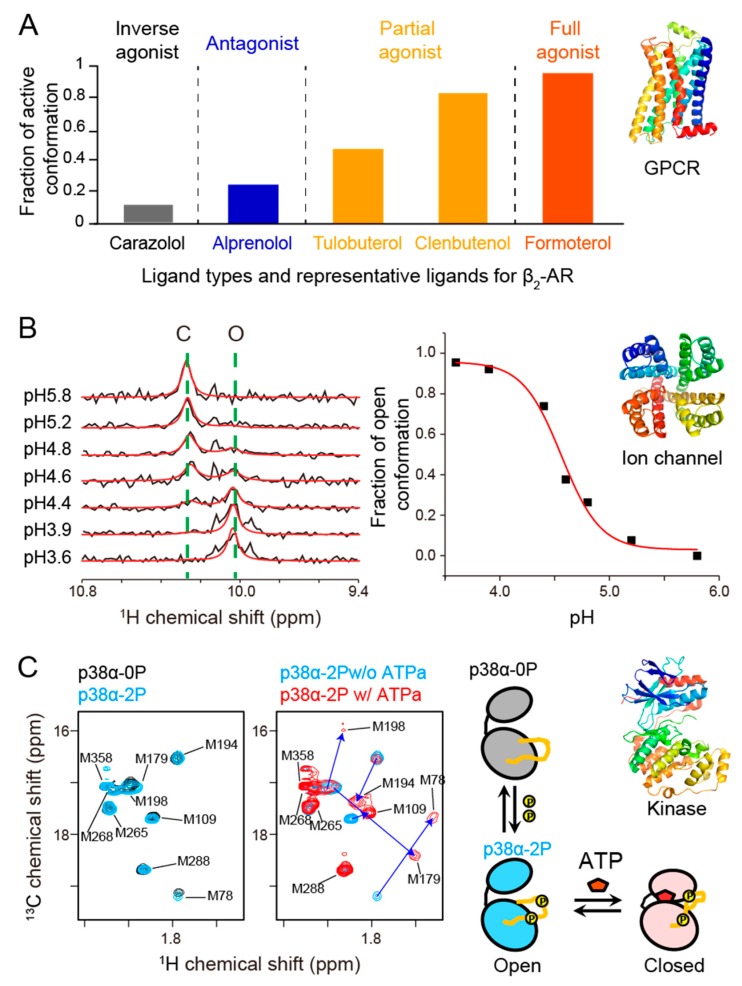
Examples of functional dynamics revealed by NMR. (**A**) Conformational equilibrium as quantified by NMR explains the efficacy of β_2_-AR agonists. The fractions of active conformation were obtained from Kofuku et al. [6]. (**B**) The conformational equilibrium revealed by NMR explains the gating properties of KcsA. Left panel: The conformational equilibrium observed in the Trp-113 signal reflects the gating of lower gate in KcsA [25]. Right panel: pH dependence of the fraction of open conformation deduced from the fitting of the Trp-113 signal. The acid dissociation constant, pKa, value of intracellular gate opening, pH 4.6, obtained from NMR, was identical to the reported pKa value obtained from electrophysiological measurements [26]. (**C**) Conformational equilibrium revealed by NMR explains the activation of kinase p38α. Left panel: The methionine methyl signals observed in the NMR spectrum indicate that dual phosphorylation is not sufficient to form the active closed conformation of p38α and that ATP is required to induce the closed conformation [27]. p38α-0P, p38α-2P, and ATPa refer to unphosphorylated p38α, dually-phosphorylated p38α, and the ATP analog, respectively. Right panel: schematic representation of the conformational state of p38α in each condition.

**Figure 3 ijms-21-01829-f003:**
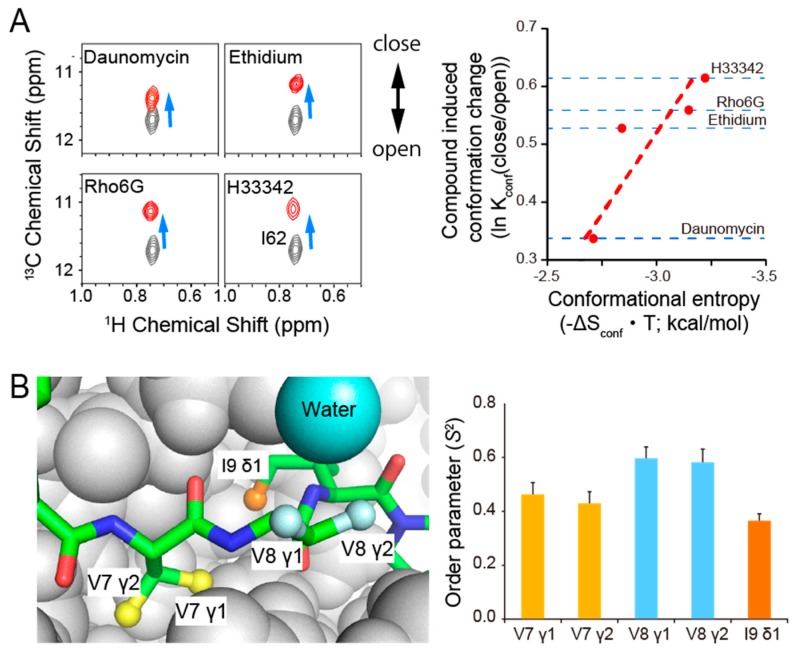
Importance of dynamics in protein–ligand interactions. (**A**) The conformational equilibrium revealed by NMR explains the binding coupled conformational entropy gain in the multidrug binding lincomycin resistance repressor LmrR. Left Panel: The chemical shift in the Ile-62 NMR signal in LmrR reflects the population of open/closed conformations in the compound binding helix^110^. Ile-62 signals from unbound and compound bound states are shown in black and red, respectively. Right panel: the population shift upon compound binding correlates with the conformational entropy gain calculated from the changes in fast-methyl dynamics (for details see reference [108]). (**B**) Conformational flexibility of a bound ligand revealed by NMR. Left Panel: The structure of the myocyte enhancer factor 2A (MEF2A) docking peptide (stick) in complex with p38α (PDB ID: 1LEW). The methyl moieties in the MEF2A peptide are shown as balls with colors corresponding to the bars in the right panel. Right Panel: Methyl order parameter (*S*^2^) values as determined by forbidden-coherence transfer (FCT) experiments (for details see reference [94]). The interface methyl moiety retains psec–nsec fast dynamics in the bound state.

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
