# Peer review of "Spotlight on the Ballet of Proteins: The Structural Dynamic Properties of Proteins Illuminated by Solution NMR"

_ijms, 2020, doi:10.3390/ijms21051829_

Round 1

Reviewer 1 Report

Tokunaga et al. provide an excellent overview of the applications of solution NMR for protein dynamics. The examples covered are very well chosen and representative of the most impactful research currently being undertaken in this area. Very well done review.

Author Response

Point to point response to Reviewer 1

Comment #1

Tokunaga et al. provide an excellent overview of the applications of solution NMR for protein dynamics. The examples covered are very well chosen and representative of the most impactful research currently being undertaken in this area. Very well-done review.

Response: We thank you the reviewer for positive comments. We also check and fix the typos.

Reviewer 2 Report

The paper reports a detailed examination of the results of several studies  about the internal mobility of proteins , membrane proteins and enzymes upon the interaction with related ligands, All the studies have been conducted by the most recent technique of the NMR spectroscopy .

The review is well written and helps very much the reader to have a overview of this theme in the field of structural biology. The number of papers  ( 132 from the pioneering BPTI experience and the results are reported concisely with high competence and well performed. Thus I have only some suggestions for a reader familial  with structural biology but less familial with NMR techniques.

Few observations:

rows 41 and 42 . the concept  of conformational entropy should be more extensively explained to a a less familial reader  as also  in the paragraph at 277 and ff.   Also to introduce to the concepts of  the entropy meter  as in ref. 106.

101  the term “excited” here and also in the following   seems to me that belongs to other spectroscopy. I believe  that it is better “ high Energy conformer “.

About Fig. 2  and related text  145 -151 : the spectroscopic evidence for identification of the close and open  states should be inserted. How it was possible to identify which is which?.

Rows 235 -237  a short description of the “relaxation dispersion measurements” for the measure of Ka should be  inserted also for the less familial readers of this important NMR technique.

Rows  181-184 I think that the limits of the extraordinary contribution of crystallography to biostructure should be  more detailed. In fact this results , very important not only masks the internal dymnamics of proteins but create salso some errors reducing all the terminals and the polar side chains to reduce their flexibilities  ( that in water solution should be free) to combine into ions couples  to allow a good  crystallization that it is often mandatory for good  X-ray diffraction results.  

A comment apart is about the title that I  found to some extent a little bit too far from the academic style; concepts like spot light and ballet seem to me to give the idea of something as a  fantastic approach   whilst the papers esamined and the results  presented are fruit of a very intensive and important  work of several NMR specialists. But this is only a simple opinion at a first view due to my aged experience in NMR. Thus the AAs are free to maintain this title if they like it.

Author Response

Point to point response to Reviewer 2

The paper reports a detailed examination of the results of several studies about the internal mobility of proteins, membrane proteins and enzymes upon the interaction with related ligands, All the studies have been conducted by the most recent technique of the NMR spectroscopy.

The review is well written and helps very much the reader to have an overview of this theme in the field of structural biology. The number of papers (132) from the pioneering BPTI experience and the results are reported concisely with high competence and well performed. Thus I have only some suggestions for a reader familial with structural biology but less familial with NMR techniques.

Response: Thank you for your valuable comments, encouragement, and suggestions. We have revised the manuscript according to your suggestions and address your concerns.

Few observations:

Comment 1: rows 41 and 42. the concept of conformational entropy should be more extensively explained to a less familial reader as also in the paragraph at 277 and ff. Also, to introduce to the concepts of the entropy meter as in ref. 106.

Response: According to the suggestion, we modified and added a concise description of conformational entropy at rows 50-57 as follows:

“Furthermore, quantitative psec to nsec dynamics measurements by NMR allow the estimation of conformational entropy, which essentially reflects the number of the accessible conformations in a macromolecule. Conformational entropy, mostly represents the amplitude of the fluctuation of bond vectors in macromolecule, which can be quantified as squared order parameters by NMR relaxation analyses. NMR-based relaxation experiments can be used to derive order parameters, Thus NMR plays a crucial role in revealing the essential contribution of the fast-conformational fluctuation of protein in regulating protein-protein affinities and allosteric regulations.”

In addition, a description of the “entropy meter” was inserted into rows 310-313 as follows:

“Although current NMR relaxation analysis techniques can typically be applied to backbone amides or methyl-bearing amino acids, the order parameters of methyl-bearing amino acids often serve as a favorable proxy of the entire order parameter including surrounding amino acids.”

Comment 2: 101: the term “excited” here and also in the following seems to me that belongs to other spectroscopy. I believe that it is better “high Energy conformer “.

Response: According to the suggestion, we substituted the term “excited” with “high energy conformer” throughout the manuscript.

Comment 3: About Fig. 2 and related text 145 -151: the spectroscopic evidence for identification of the close and open states should be inserted. How it was possible to identify which is which?

Response: Thank you for the critical comment. First of all, we reworded the nomenclatures as cytosol- and matrix-facing states, because the membrane protein, AAC, described here, resides in the inner mitochondrial membrane. Under these new definitions, the dominant state was assigned as a cytosol-facing state, because the 1H-15N HSQC spectrum of the apo form matched the spectrum of the inhibitor-bound state, where the crystal structure had been solved, which represented the cytosolic-facing state. The lowly populated state was assumed to be the matrix-facing state, because the relaxation dispersion experiments indicated the residues corresponding to cytosol-facing part of the protein showed conformational exchange. The exchange rate between the two conformations derived from relaxation dispersion experiments matches the exchange rate postulated for the transport which adds further evidence to our assumption that minor state in the matrix-facing state. Taking these things into consideration, we rewrote the description at rows 160-168 as follows and added a reference for the X-ray structure paper.

“For example, in yeast ADP/ATP carrier 3 (AAC), a protein that is found in the inner mitochondrial membrane, the 1H-15N HSQC-TROSY spectrum shows that AAC in solution dominantly resides in the cytosol-facing conformation. This is conformation is similar to the X-ray structure of the inhibitor-bound state of AAC1. However, NMR relaxation dispersion experiments reveal that there is another conformational state populating ~2%. The minor state is assumed to be related to the matrix-facing conformation, since residues that show conformational exchange are localized to the matrix-facing side. In this system, the presence of substrate accelerates the exchange between two conformations, which might contribute to its fast transport.

  1. Robinson, A. J.; Kunji, E. R. S., Mitochondrial carriers in the cytoplasmic state have a common substrate binding site. Proceedings of the National Academy of Sciences of the United States of America 2006, 103 (8), 2617.

Comment 4: Rows 235 -237: a short description of the “relaxation dispersion measurements” for the measure of Ka should be inserted also for the less familial readers of this important NMR technique.

Response: Thank you for the critical comment. We need to clarify that the “relaxation dispersion (RD) measurements” were not used to estimate Ka (or Ki) in this study. Instead the RD measurements were used to estimate rate of conformational switching from ground to excited conformations in each compound bound state. We apologize for the confusion. What we intended to point out here is that the rate of conformational changes correlate with Ki and koff for this system, thus, internal protein motion might be critical for ligand dissociation. We revised the text as follows in rows 259-263 and also added an introductory description of relaxation dispersion measurements in rows 41-50.

Rows 41-50

“For this purpose, relaxation dispersion measurements give quantitative information about relatively slow (μsec to msec) timescale equilibrium. In the Carr-Purcell-Meiboom-Gill (CPMG) type relaxation dispersion experiment, inversion pulses are applied at an array of frequencies during the fixed transverse relaxation period. A residue that does not undergo conformational exchange will experience one chemical shift and corresponding intensity will not depend on the CPMG frequency. However, the chemical shift of a residue that experiences conformational exchange will oscillate between the various conformers and hence the intensity of the corresponding resonance will depend on the CMPG frequency. The effective transverse relaxation rate measured as a function of pulsing frequency harbors information about the exchange rate, the population of the states, and the chemical shift difference between the states.

Rows 259-263

CPMG relaxation dispersion measurements showed that the rates of ms-timescale internal motions in the enzyme in each compound bound state correlates with the inhibitory constant (Ki) and koff for antifolates. This suggest that internal protein motion is critical for ligand dissociation by mechanically initiating the ligand dissociation.

Comment 5: Rows 181-184:I think that the limits of the extraordinary contribution of crystallography to biostructure should be more detailed. In fact this results, very important not only masks the internal dymnamics of proteins but creates also some errors reducing all the terminals and the polar side chains to reduce their flexibilities (that in water solution should be free) to combine into ions couples to allow a good crystallization that it is often mandatory for good X-ray diffraction results. 

Response: We thank to the comment. We added a description of structural perturbations potentially caused by crystallization that can mislead into the inappropriate understanding of mechanisms of kinase function into rows 200-208, as follows:

“Although X-ray crystallography has been essential to define the structural basis of protein kinase functions, characterizing functionally important dynamic can sometimes be lost by crystallization. A conformational equilibrium, such as an inter-lobe open/close equilibrium, can be altered to choose the conformation that fits better to the crystal lattice. In addition, a flexible loop, as exemplified by the activation loop in kinases, could also artificially rigidified by crystal packing, which masks the modular role of the loop in regulating the activity. Furthermore, the indicator of local dynamics in crystal structures, B-factors, can underestimate the dynamics of surface-exposed polar side chains, due to the condensation-enhanced attraction to ions that are supplemented for high quality crystals.”

Comment 6: A comment apart is about the title that I found to some extent a little bit too far from the academic style; concepts like spot light and ballet seem to me to give the idea of something as a fantastic approach whilst the papers examined and the results presented are fruit of a very intensive and important work of several NMR specialists. But this is only a simple opinion at a first view due to my aged experience in NMR. Thus the AAs are free to maintain this title if they like it.

Response: Thanks for the thoughtful opinion. We realized that the title is a bit casual, and this was done with intent. The idea here is to emphasize the strength of NMR in specifically characterizing, or spotlighting, the exciting and beautiful dynamics of proteins in function, to which we refer as a ballet. The title was framed with the intent that a non-NMR person would be encouraged to read the article.